# Characterization of 3D-Bioprinted In Vitro Lung Cancer Models Using RNA-Sequencing Techniques

**DOI:** 10.3390/bioengineering10060667

**Published:** 2023-06-01

**Authors:** Sheng Zou, Jiayue Ye, Yiping Wei, Jianjun Xu

**Affiliations:** The Second Affiliated Hospital of Nanchang University, Nanchang 330030, China; 361439920078@email.ncu.edu.cn (S.Z.); 401441620028@email.ncu.edu.cn (J.Y.); weiyip2000@hotmail.com (Y.W.)

**Keywords:** lung cancer, 3D bioprinting, in vitro model of lung cancer, hydrogel, RNA-seq

## Abstract

Objective: To construct an in vitro lung cancer model using 3D bioprinting and evaluate the feasibility of the model. Transcriptome sequencing was used to compare the differential genes and functions of 2D and 3D lung cancer cells. Methods: 1. A549 cells were mixed with sodium alginate/gelatine/fibrinogen as 3D-printed biological ink to construct a hydrogel scaffold for the in vitro model of lung cancer; 2. A hydrogel scaffold was printed using a extrusion 3D bioprinter; 3. The printed lung cancer model was evaluated in vitro; and 4. A549 cells cultured in 2D and 3D tumour models in vitro were collected, and RNA-seq conducted bioinformatics analysis. Results: 1. The in vitro lung cancer model printed using 3D-bioprinting technology was a porous microstructure model, suitable for the survival of A549 cells. Compared with the 2D cell-line model, the 3D model is closer to the fundamental human growth environment; 2. There was no significant difference in cell survival rate between the 2D and 3D groups; 3. In the cell proliferation rate measurement, it was found that the cells in the 2D group had a speedy growth rate in the first five days, but after five days, the growth rate slowed down. Cell proliferation showed a declining process after the ninth day of cell culture. However, cells in the 3D group showed a slow growth process at the beginning, and the growth rate reached a peak on the 12th day. Then, the growth rate showed a downward trend; and 4. RNA-seq compared A549 cells from 2D and 3D lung cancer models. A total of 3112 genes were differentially expressed, including 1189 up-regulated and 1923 down-regulated genes, with *p*-value ≤ 0.05 and |Log2Ratio| ≥ 1 as screening conditions. After functional enrichment analysis of differential genes, these differential genes affect the biological regulation of A549 cells, thus promoting lung cancer progression. Conclusion: This study uses 3D-bioprinting technology to construct a tumour model of lung cancer that can grow sustainably in vitro. Three-dimensional bioprinting may provide a new research platform for studying the lung cancer TME mechanism and anticancer drug screening.

## 1. Introduction

Lung cancer ranks second in morbidity and first in mortality among all malignancies [1]. Surgery and chemotherapy are the main cancer treatments. In recent years, stereotactic ablation of radiation therapy and targeted drug therapy have also progressed [2,3,4,5]. However, the 5-year survival rate of lung cancer remains low. In the study of lung cancer, 2D cell-line models and patient-derived xenografts (PDXs) have been regarded as the gold standard [6]. The 2D cell-line model is very useful in gene research because it is simple, fast, and cheap [7,8,9,10]. The 2D cell-line model grew infinitely in a Petri dish towards this plane, and the proliferation rate was related to the size of the 2D culture area. However, 2D cell lines can only represent a tumour cell subpopulation, which lacks the conditions of the original human microenvironment in the growth environment, and it is difficult to maintain some functional biological characteristics of the original tumour in the proliferation process [11,12,13]. In addition, 2D cell-line models cannot simulate the microenvironment of 3D tissues [14,15]. The PDX model can more truly model the human tumour microenvironment in the experiment to ensure cells’ gene expression and phenotype [16,17]. However, due to the low success rate of the PDX model in the modelling process and a large number of expenses and time costs, use of the PDX model is also limited [18]. Moreover, animal models do not respond to predicting human toxicology and pathology [19].

3D-bioprinting technology has attracted much attention due to its great potential in tissue engineering production [20]. Many research groups have found that 3D bioprinting has many advantages over traditional biological tissue engineering, such as exact control of cells, high resolution of cell observation, low cost, location and distribution of materials, the ability to personalize the construction of tissue engineering products, and maximum imitation and minimum displacement. Moreover, it solves the problem of restricting mass production and has advantages in terms of stability, manufacturing speed, and cost [21,22,23,24,25,26]. There are a variety of 3D-bioprinting technologies, including extrusion, jetting, and vat-photopolymerization-based bioprinting. Three-dimensional extrusion bioprinting combines fluid distribution and automated robotic systems [27]. During the printing process, the cells are precisely deposited in three-dimensional space by computer control, while the continuous deposition characteristics of the filament provide better institutional integrity [27]. Jetting bioprinting is accomplished by the non-contact injection of nano-litre cell-laden droplets. In the printing process, the distance between the nozzle and the matrix is controlled to control the impact speed of the droplet and droplet evaporation to maintain high cell viability and proliferation rate of the printed cells [28]. Vat photopolymerization bioprinting uses a computer-controlled laser beam to scan the surface of a photopolymerization bio resin along a predetermined pattern, to solidify and adhere the photopolymerization resin to the previous layer, and to build the platform. Then, it continues to apply the photopolymerization resin, and so on, to achieve a complex 3D structure [29,30]. Three-dimensional-bioprinted tissue models can replace 2D models and animal models [31]. Yu Shrike Zhang et al. [32] established complex and accurate models in vitro to simulate the tumour microenvironment (TME) through 3D-bioprinting technology, which can conveniently study the occurrence and development of cancer. Three-dimensional bioprinting, which controls the distribution of cells, active molecules, and biomaterials, yields the printing of different tumour microenvironment cells and reproducing times, paving the way for innovative platform technologies for in vitro tumour research [33,34]. Nowadays, there are many kinds of 3D in vitro models, such as spheroid cultures, biopolymer scaffolds, cancer-on-a-chip devices, and ex vivo tissue slices; extrusion 3D bioprinting, however, has a competitive advantage because of its ability to precisely control and define the desired structure and location of multiple cell types in a high-throughput manner, simulate the application of the TME, and be used as a pre-screening tool in immunotherapy [35,36]. Three-dimensional bioprinting provides cell–cell and cell–matrix interactions by mimicking the 3D heterogeneity of real tumours. It has potential applications as a personalized in vitro model for anticancer drug screening and the establishment of precision therapies [37]. Zhao Yu et al. [38] used gelatine/alginate/fibrinogen as a hydrogel and established a model of cervical cancer in vitro as a research platform. Farben Meng et al. [39] established a natural tumour microenvironment using A549 cells to construct in vitro metastasis models through 3D-bioprinting technology. This research provides a proof-of-concept platform for anticancer drug research, development, and therapeutic methods.

Cellular bio-ink materials are very important in 3D bioprinting. The morphology, cell activity, proliferation, migration, and aggregation of cells in the printed hydrogel scaffolds largely depend on bio-ink characteristics [40,41,42]. Sodium alginate is a natural anionic polysaccharide derived from brown algae and can be used as a natural polymer for preparing hydrogels [43]. Alginate molecules can interact with bivalent cations (such as Ca^2+^, Sr^2+^, and Ba^2+^) and bind to glucuronic acid to complete chemical crosslinking, and the process is spatially linked to form a hydrogel scaffold [44]. Sodium alginate can crosslink with CaCl_2_ to form an elastic hydrogel scaffold. The crosslinking scaffold has strong mechanical properties, high stability, and good biocompatibility and does not affect the growth vigour of cells in the scaffold during the crosslinking process [45]. Gelatine is a biodegradable polypeptide, derived mainly from the hydrolysis of collagen, which is widely found in the body’s connective tissues [46]. Gelatine has many clinical uses, unique biocompatibility, rapid degradation, and non-antigenic properties. In 3D-bioprinting technology, sodium alginate/gelatine hydrogel scaffolds have been widely used in disease modelling, high-throughput drug screening, and bioartificial apparatus manufacturing [47,48,49,50,51]. Sodium alginate and gelatine have been developed in medical research for many years. Studies have proved that sodium alginate/gelatine has printability and biocompatibility when printing various living cells [40,52].

The mature functional lung is mainly composed of the trachea, gas-exchange acinus (there is an air–blood barrier in the acinus, mainly composed of human lung epithelial cells, human endothelial cells, and human lung fibre cells), effective vascular-circulation system, and lung surface active system to promote lung expansion [53,54]. The anatomy of a mature-functioning lung is quite complex, making it a challenge to replicate lung models. The average thickness of the alveolar sacs in exchange for gas is only 0.5 μm. There has yet to be a real 3D lung model in vitro. Lenke Horvath et al. [55] used jetting 3D-bioprinting technology to prepare similar substances of the human air–blood barrier composed of endothelial cells, basement membrane, and epithelial cell layers. Ngwei Long et al. [56] prepared an in vitro alveolar model by the composition of human lung epithelial cells, human endothelial cells, and human lung fibrocytes. In addition, Dongeun et al. [57] designed a micro-physiological organoid with alveolar function on the chip to simulate the human lung capillary interface. In this study, a lung cancer model was prepared using A549 cells by 3D-extrusion-bioprinting technology. The model did not consider the function of lung gas exchange, and the study focused on tumour cell growth regulated by the tumour microenvironment in the lung cancer model to explore further the feasibility and biological evaluation of the in vitro tumour model of lung cancer. Three-dimensional-bioprinting technology can be used to simulate the 3D environment of human tumours with the advantages of low cost and short modelling time.

## 2. Materials and Methods

### 2.1. Cell Culture

Human non-small lung cancer A549 cells were purchased from the Chinese Academy of Sciences in Shanghai. The culture medium was cultured with complete medium DMEM in an incubator of 37 °C and 5% CO_2_, and the medium was changed once every 2–3 days. When the cells were cultured to about 90%, the digestive passage was performed using 0.25% trypsin.

### 2.2. Materials

The weighed sodium alginate and gelatine powder were added into 0.9% normal saline and fully dissolved and mixed in a water bath at 60 degrees Celsius so that the sodium alginate and gelatine concentrations reached 4% and 20%, respectively. At the same time, 0.9% normal saline was used to configure 3% fibrinogen. The sterilized 4% sodium alginate, 20% gelatine, and 3% fibrinogen were mixed in a ratio of 1:1:1 and dissolved in a 37 °C water bath for later use. 

### 2.3. D Bioprinting

The pre-cooled sodium alginate/gelatine/fibrinogen mixture containing cell suspension was first transferred into the syringe and then installed into the sleeve core of the Z-axis of the 3D bioprinter. The 3D bioprinter is shown in Figure 1. The Petri dish was placed on the printing platform with variable temperatures, the printing height was adjusted, and the printing route was set according to the program. After printing, the sterile 3% CaCl_2_ was crosslinked for 1 min, and the hydrogel fibres were sucked out after the stable state was observed. For the excess CaCl_2_ solution, the printed biological scaffold was cleaned with normal saline three times, then 20 μL/mL of thrombin crosslinked fibrinogen was added for 30 min. TGaes auto crosslinks gelatine in the hydrogels in the cell suspension. After crosslinking, normal saline was used for cleaning three times, and then the complete medium was added and cultured in an incubator at 37 °C and 5% CO_2_. The medium was replaced about two days later, and regular observations were made. The 3D-bioprinting process is shown in Figure 2A. After printing, the sterile 3% CaCl_2_ was crosslinked for 1 min, and the hydrogel fibres were sucked out after the stable state was observed. For excess CaCl_2_ solution, the printed biological scaffold was cleaned with normal saline three times, then 20 μL/mL of thrombin crosslinked fibrinogen was added for 30 min. TGaes auto crosslinks gelatine in hydrogels in the cell suspension. After crosslinking, normal saline was used for cleaning three times, and then the complete medium was added and cultured in an incubator at 37 °C and 5% CO_2_. The medium was replaced about two days later, and regular observations were made. The crosslinking process is shown in Figure 2B.

### 2.4. Alginate Shell Removal

Ethylene diamine tetraacetic Acid (EDTA) and sodium citrate were mixed at a ratio of 1:2, then dissolved in normal saline and sterilized at a high temperature for reserve use. After forming the 3D-printed hydrogel, A549 cells grew in the hydrogel scaffold for a certain period, and then EDTA and sodium citrate were used to remove the sodium alginate shell. A549 cells grown in the 3D in vitro model could be obtained for subsequent experimental studies.

### 2.5. Scanning Electron Microscopy

The printed hydrogel scaffold was soaked in a 3% glutaraldehyde solution to fix. Ketone and isoamyl acetate were mixed at a ratio of 1:1 and added into the hydrogel scaffold for soaking. The hydrogel scaffold was removed, and acetonitrile gradient dehydration was carried out. The dehydrated hydrogel scaffold was vacuum-dried and dehydrated. A layer of platinum was sprayed on the sample’s surface using the vacuum jetting method and then photographed using an electron microscope.

### 2.6. Live/Dead Assay

The 3D-printed hydrogel scaffold was placed in the 24-well plate, and 100 μL of the mixed working liquid was added to the hydrogel scaffold to be dyed for incubation. A total of 5 μL calcium AM and 5 μL propidium iodide were mixed into 10 mL PBS. The 3D-printed hydrogel scaffold was placed in the 24-well plate, and 100 μL of the mixed working liquid was added to the hydrogel scaffold to be dyed for incubation. The dyeing solution in the incubated 3D hydrogel scaffold was sucked out and cleaned with PBS. The dyed hydrogels were observed under a fluorescence microscope. Five visual fields were randomly selected for each sample, and the photos were repeated three times.

### 2.7. Cell Proliferation Assay

A 3D-printed hydrogel scaffold containing cells was placed into a 24-well plate, and the Alamar Blue stock solution was diluted using complete medium (1:9 = Alamar Blue:complete medium). A total of 1 mL working solution was added to each well, and the blank control group was incubated at 37 °C and 5% CO_2_ for 2 h in the dark. After incubation, the work was moved to a 96-well plate at the rate of 100 μL, and three multiple wells were set. The 570 nm and 600 nm absorbance were calculated using the enzyme marker and recorded. The hydrogel in the 24-well plate was cleaned with normal saline to remove the remaining working liquid, and the complete medium was replaced to continue the culture. OD values of the 3D-bioprinted hydrogel were recorded from the first day of printing and recorded every three days. Finally, the data was standardized until the first day and collected and mapped statistically.

### 2.8. Paraffin-Embedded and Pathological Staining

The 3D-bioprinted hydrogels were fixed with 4% paraformaldehyde and embedded in paraffin. The setting mechanism created a 3 μm thick section, and H&E staining was performed according to routine histology.

### 2.9. RNA Sequencing and Bioinformatics Analysis

After RNA extraction and RNA sequencing, DESeq2R software was used for different analyses. Cluster Profiler software was used for GO enrichment analysis and KEGG pathway enrichment analysis of differential genes. The enrichment analysis examined the relationship between the differential genes and biological functions and signalling pathways by comparing each group’s up-regulated and down-regulated differential gene sets. GSEA analysed the phenotypic differences between 2D and 3D differential genes.

### 2.10. Statistical Analysis

Each experiment was repeated three times, and the results were expressed as mean ± SD. *t*-test was used to compare the two groups, and *p* < 0.05 was statistically significant.

## 3. Results

### 3.1. 3D-Printed Models of Lung Cancer Tumours In Vitro

To construct a 3D in vitro tumour model of lung cancer, a hydrogel scaffold containing A549 was extruded using a threaded pump built into the 3D bioprinter. The sodium alginate/gelatine hydrogels were temperature-sensitive hydrogels. Under a low-temperature environment, hydrogels could be printed more easily, which was beneficial to increase the success rate of printing. In addition, at low temperatures, the metabolic rate of the cells in the hydrogel decreased, helping to increase the survival rate of the cells after printing. According to the pre-set printing parameters of the 3D printer, the bio-ink was extruded and printed into a grid hydrogel on the scaffold (Figure 3D). The contact distance between the cellular hydrogel scaffold and the medium was the same in the grid state. After completing the crosslinking between the cellular sodium alginate/gelatine/fibrinogen hydrogel scaffold and CaCl_2_/TGesa/thrombin crosslinking agent, the complete medium can permeate the hydrogel scaffold freely. The cells in the hydrogel scaffold could freely absorb all the nutrients in the medium and release metabolites.

### 3.2. Cell Survival Analysis

A549 showed good cell activity when cultured in a 3D-printed hydrogel scaffold, and the survival rate of cells in the hydrogel scaffold was tested with live/dead reagents. On the 15th day of culture, the survival rate of A549 cells in the 2D group was 91.10% ± 1.3%, and that in the lung cancer model in vitro was 89.76% ± 2.3%. There was no significant difference in cell survival rate between 2D and 3D groups, and *p* < 0.05 was not statistically significant (Figure 3A–C).

### 3.3. Cell Proliferation Capacity Analysis

In 2D culture, the cells grew on a single flat-surface layer. The cell growth rate was very fast within five days. Cell proliferation is stable when the cell growth density reaches more than 90%. However, in a 3D culture environment, cells grow in all directions in a 3D space, forming a tumour ball in a limited space. Cells were always in a slow growth state at the beginning, and then the basic growth of cells stabilized. At the same time, Alamar Blue was used every three days to determine the OD values of cell proliferation in 2D and 3D culture environments, and quantitative analysis was made (Figure 3E).

After the OD value was determined by Alamar Blue and quantitative analysis was conducted, it was found that the growth rate of the lower cells cultured in 2D was very fast in the first five days. After five days, cell growth slowed down due to limited flat space, while after nine days, cell proliferation showed a decline. However, in the cultural environment of the 3D-printed hydrogel, different from the 2D culture environment, cells grew slowly at the beginning. When the 3D environment space of cells was insufficient, the growth rate slowed down and stayed stable. Cultured in a 3D hydrogel environment, the growth rate peaked before 12 days, and then the growth rate showed a downward trend (see 3D hydrogel inside A549 cell growth (Figure 3F–M)).

### 3.4. Scanning Electron Microscope Observation

The hydrogel was observed under an electron microscope, and tumour balls with protrusion could be observed. After the lyophilized hydrogel, the epidermis of some tumour balls was damaged, and tumour cells growing and increasing inside the tumour balls could be observed (Figure 4A–F).

### 3.5. Paraffin Section and H&E Staining

After the fifth day of the culture of the 3D-printed hydrogel, H&E staining was carried out, and single growing cells could be found in each hole. After day 15 of the culture, it was observed that the cells in each pore grew and proliferated into multiple cells, and the hydrogel space occupied by the cells became larger, similar to the growth pattern of tumours in vivo (Figure 4G–J).

### 3.6. Difference Analysis

Through DESeq2R software, genes in our library were analysed, and the filter condition was *p* ≤ 0.05 or |Log_2_Ratio| ≥ 1. Genes meeting the screening criteria were defined as differential genes, log_2_foldchange > 0 indicates that the gene is up-regulated, and log_2_foldchange < 0 indicates that the gene is down-regulated. Compared with the 2D group, there were 3112 differentially expressed genes in the 3D group, including 1189 up-regulated genes and 1923 down-regulated genes. We visualized all the different genes in a volcano map (Figure 5A). At the same time, the hierarchical clustering method was used for cluster analysis and visualization as a heat map (Figure 5B). It is suggested that 2D and 3D growth environments could change the difference in gene expression in lung cancer cells.

### 3.7. GO Functional Enrichment Analysis

The top 30 items of differential gene enrichment were prepared into bar charts; *p* < 0.05 represents significant enrichment (Figure 6).

### 3.8. KEGG Path Enrichment Analysis

KEGG pathway enrichment analysis was used to collect differential genes and compare the genes in signal pathways in the database to infer the signal pathways in which differential genes participate. *p* < 0.05 represents significant enrichment of differential gene signalling pathways, and the results of significantly enriched pathways were presented as scatter plots (Figure 7). The up-regulated KEGG enrichment pathways include the PI3K-Akt signalling pathway, iron death signalling pathway, etc. The down-regulated KEGG enrichment pathway included the calcium signalling and cell-adhesion molecule.

### 3.9. GSEA Enrichment Analysis

GSEA analysis showed that the growth of A549 cells in the 3D group was related to cell-cycle signalling pathways, such as NF-KAPPA-B and PI3K-Akt (Figure 8). Three-dimensional environment growth enhanced the malignant degree of lung cancer cells and promoted the development of lung cancer cells.

## 4. Discussion

3D-bioprinting technology is an extension of 3D-printing technology, using biocompatible materials and cells to build biological structures that mimic the complex structure and function of the microenvironment and matrix components in the body. Many research groups have demonstrated that a 3D culture environment can promote the activity, dryness, and migration of tumour cells and promote the progression of tumour cells [58,59,60]. Our group observed that, during the construction of 3D in vitro models, lung cancer cells maintain high viability. Thomas M Keenan [61] et al. proposed that the concentration gradient of biomolecules plays a vital role in chemotaxis and tumour metastasis. The spheroids formed by hydrogels in this study form a concentration gradient of nutrients and oxygen, which more closely simulates the nutrient gradient in human tumours. Chen et al. [62] used a double-crosslinked model (Ca^2+^ to crosslink alginate molecules and transglutaminase (TG) to crosslink gelatine molecules). Printed hydrogels have improved the physicochemical properties of molecules, such as water retention, hardness, and structural integrity, as well as biochemical properties, such as cytocompatibility. In this study, fibrinogen and thrombin were added to the hydrogel scaffold under the premise of a double-crosslinked model, which made the hydrogel scaffold more stable and ensured cell activity, and the survival rate of cells in the scaffold reached 89.76 ± 2.3%. Gao et al. [7] found that bimolecular green fluorescent protein could freely diffuse in 4% sodium alginate fibre. In this study, the 4% sodium alginate shell, after crosslinking, had good structural integrity and permeability, allowing the shell to diffuse cytokines and nutrients freely.

Lung cancer cells in the 2D model initially grew faster than those in the 3D model. Then, there was limited room for cells to grow in 2D. When the dish is filled with cells, cell proliferation decreases, and cells die in sheets. In the 3D model, the cells increased slowly but steadily. The growth rate slowly decreased around 12 days. In a three-dimensional environment, the cell’s environment is in three-dimensional space, and it can grow around it and form a spherical cell. Zhao et al. [38] used the same method as in this study to print Hela cells for a cervical tumour model in vitro. The 3D environment showed a higher proliferation rate and tended to form cellular spheroids. This is consistent with the growth of the in vitro model constructed in this experiment. In our study, we found that tumour cells can grow around 3D space, and after some time, they are confined by hydrogels and then stop growing, forming tumour cell aggregates. In the study model, the growth of A549 cells is very similar to the growth of human tumours. After tumour cells grow to a certain period, they have a strong proliferation rate and invasion ability, which can invade and squeeze surrounding tissues.

In this study, we did not conduct more in-depth research on the mechanism but compared the differences in gene expression between the 2D and 3D groups of cells through RNA-seq research and performed functional enrichment screening. The results showed that the changes in the main signalling pathways of lung cancer cells cultured in the 3D environment were related to the cell cycle, PI3K-Akt, NF-κB, and other classical signalling pathways. Multiple signalling pathways are involved in the development of lung cancer. The relationship between lung cancer and the NF-κB signalling pathway has been studied for many years, and many studies have suggested that the NF-κB signalling pathway is involved in the progression of lung cancer [63,64,65,66]. Manish Kumar [67] et al. used modified hanging-droplet method technology to construct 3D multicellular sphere culture. He found that in a 3D cell culture system, lung cancer cells are highly invasive by inducing epithelial-to-mesenchymal transition (EMT), leading to the activation of NF-κB. Compared with the in vitro model created in this experiment, the 3D culture system created by hanging droplets is much rougher. The mechanism of lung cancer development in the 3D microenvironment has been further studied, which also verifies that lung cancer cells can promote the progression of lung cancer cells by activating the NK-KB signalling pathway in the 3D environment. Yuji Sakuma [68] et al. also found that differences between 2D and 3D microenvironments lead to increased activation of NF-κB and induce cell immune apoptosis, thus making lung cancer cells more active in a 3D environment. Li Ying et al. [69] found the inhibitory effect of PI3K on A549 apoptosis through 3D microfluidic chip technology. All the above studies conducted a deeper study on the changes of cancer cell signalling pathways in 3D culture and verify this experiment’s results through functional enrichment of differential genes.

In this study, a three-dimensional space for the growth of A549 cells was constructed by 3D-bioprinting technology, preliminarily completing a space field that simulates the growth environment of the human body. Meng Fanben et al. [39] established an A549 cell transfer model in vitro using 3D-bioprinting technology and a functional vasculature system, and they established a suitable drug-screening platform. The experiment highlights the advantages of 3D tumour models in simulating the target cell microenvironment, which provide new ideas for diagnosing and treating clinical patients and for our research team to provide further research ideas. Zhiyun Xu et al. [70] designed a 3D co-culture sensitivity test platform of microfluidic chips to test the drug sensitivity of lung cancer cells in 3D culture, conduct targeted drug resistance tests for individual patients, and experiment with the personalized precision therapy of lung cancer. The 3D microfluidic chip technology in this study is an ideal research platform for signalling pathway research and drug-screening research compared to the in vitro model in this study. However, regarding the difficulty of platform construction and cost savings, the in vitro lung cancer model we designed has more advantages. In clinical practice, 3D tumour models in vitro serve as a new platform on which targeted drug resistance tests can be carried out for individual patients to experiment with the personalized and precise treatment of lung cancer.

## 5. Conclusions

This study uses 3D-bioprinting technology to construct a lung cancer tumour model that can grow sustainably in vitro. Three-dimensional bioprinting may provide a new research platform for studying the mechanism of the lung cancer tumour microenvironment and anticancer drug screening.

## Figures and Tables

**Figure 1 bioengineering-10-00667-f001:**
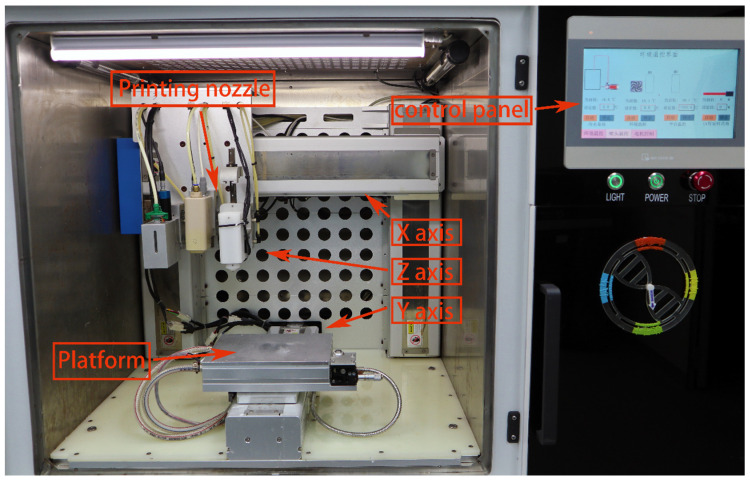
The appearance of a 3D bioprinter with multiple sprinkler heads.

**Figure 2 bioengineering-10-00667-f002:**
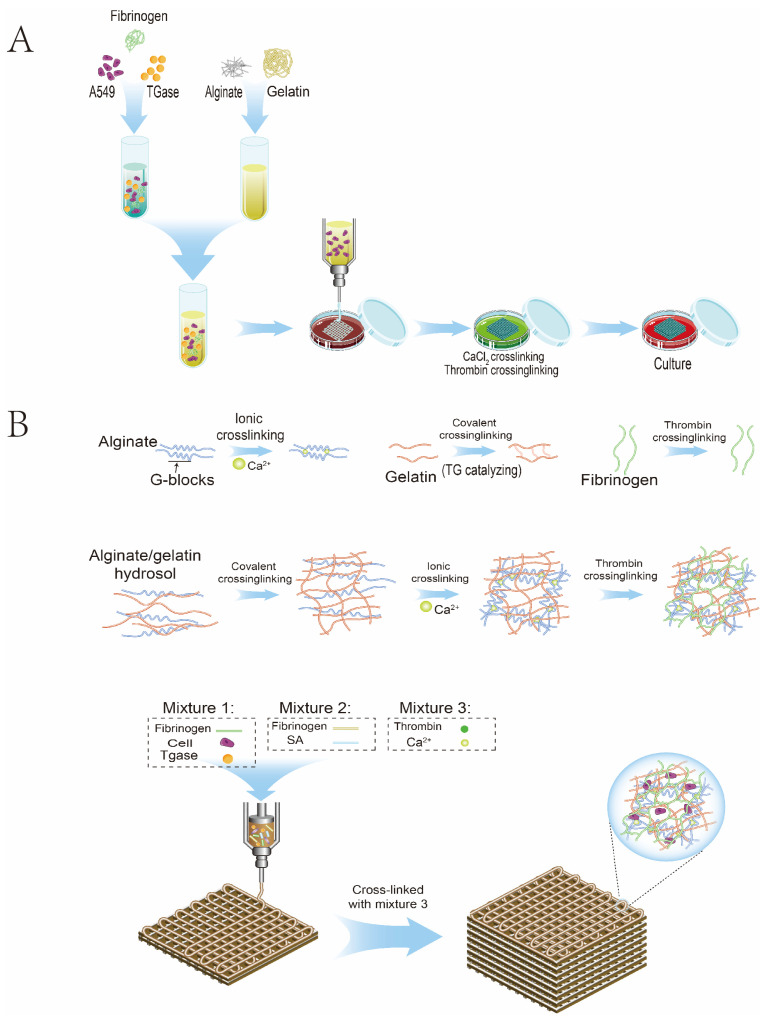
Schematic diagram of 3D bioprinting. (**A**) Flowchart of the 3D-bioprinting process. (**B**) Hydrogel crosslinking diagram.

**Figure 3 bioengineering-10-00667-f003:**
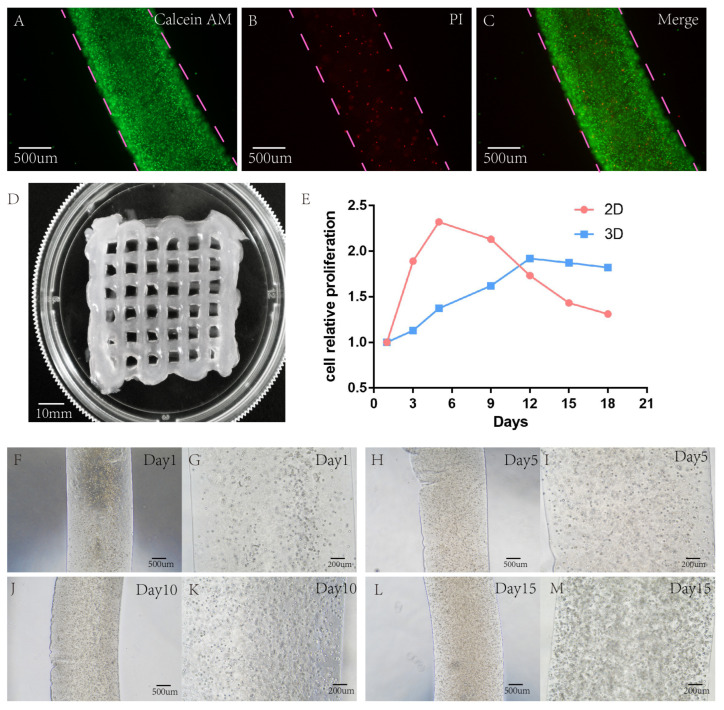
Detection of live/dead cells of the hydrogel scaffold containing A549 after 15 years of growth in 3D-bioprinted hydrogel. (**A**–**C**) Green represents the live cells stained with calcein, and red represents the dead cells stained with PI. (**D**) Successful printing of a gridded hydrogel scaffold containing cells. (**E**) Comparison of cell proliferation rate between 2D and 3D model. (**F**–**M**) Growth of cellular hydrogel under a light microscope at days 1, 5, 10, and 15. Scale: A, B, C, F, H, J, L 500 μm; G, I, K, M 200 μm.

**Figure 4 bioengineering-10-00667-f004:**
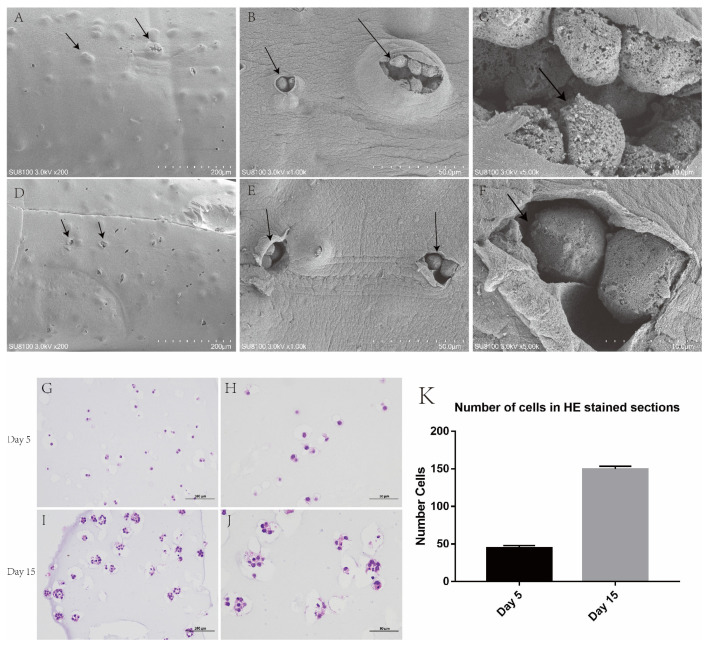
Bulging tumour cell aggregates were seen in hydrogel under an electron microscope and H&E staining. (**A**–**F**) An enlarged tumour cell aggregate was seen under the hydrogel. (**G**,**H**) Day 5 of cell hydrogel scaffold growth. (**I**,**J**) Day 15 of cell hydrogel scaffold growth. (**K**) Number of cells in H&E-stained sections. Scale: A, E: 200 μm, H, I: 100 μm, B, F, J, K: 50 μm, C, G: 10 μm.

**Figure 5 bioengineering-10-00667-f005:**
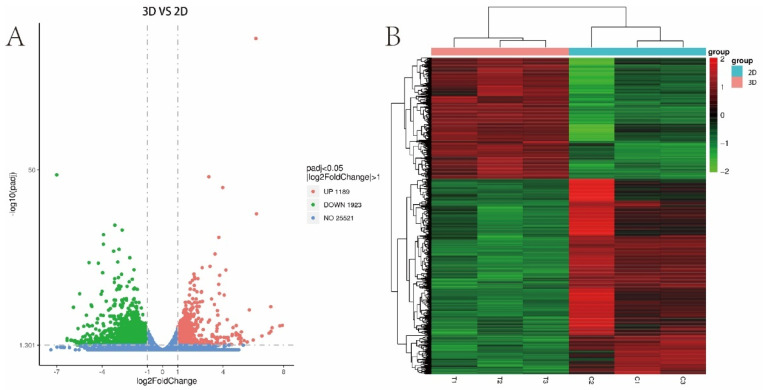
Differential genes. (**A**) Volcano maps of A549 cell differential genes in 2D and 3D groups. (**B**) Heat maps of 2D and 3D A549 cell differential genes for cluster analysis.

**Figure 6 bioengineering-10-00667-f006:**
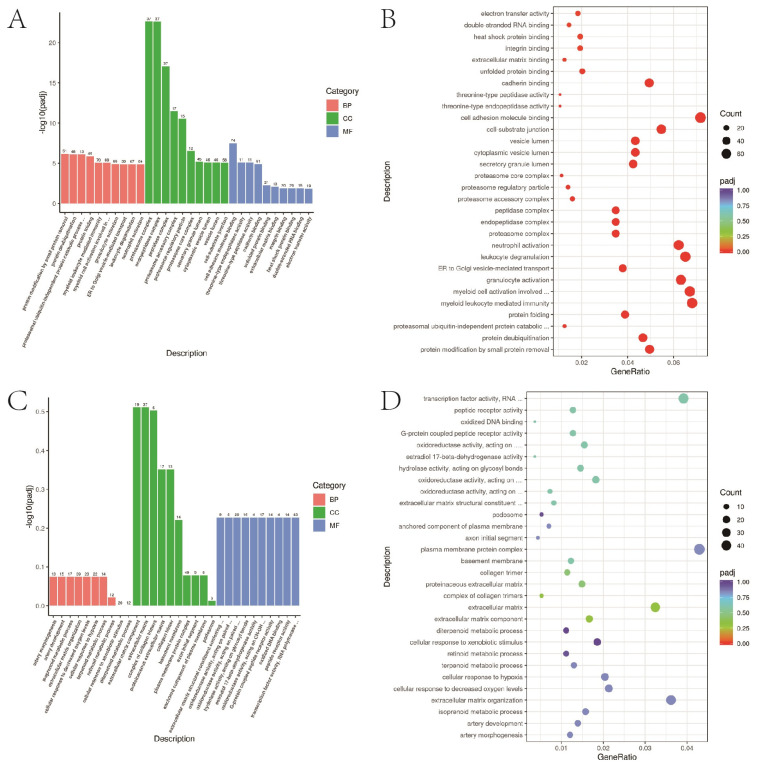
GO functional enrichment. (**A**,**B**) A549 cells in 2D and 3D groups up-regulated the first 30 items of GO functional enrichment of differential genes. (**C**,**D**) A549 cells in 2D and 3D groups down-regulated the top 30 items of GO functional enrichment.

**Figure 7 bioengineering-10-00667-f007:**
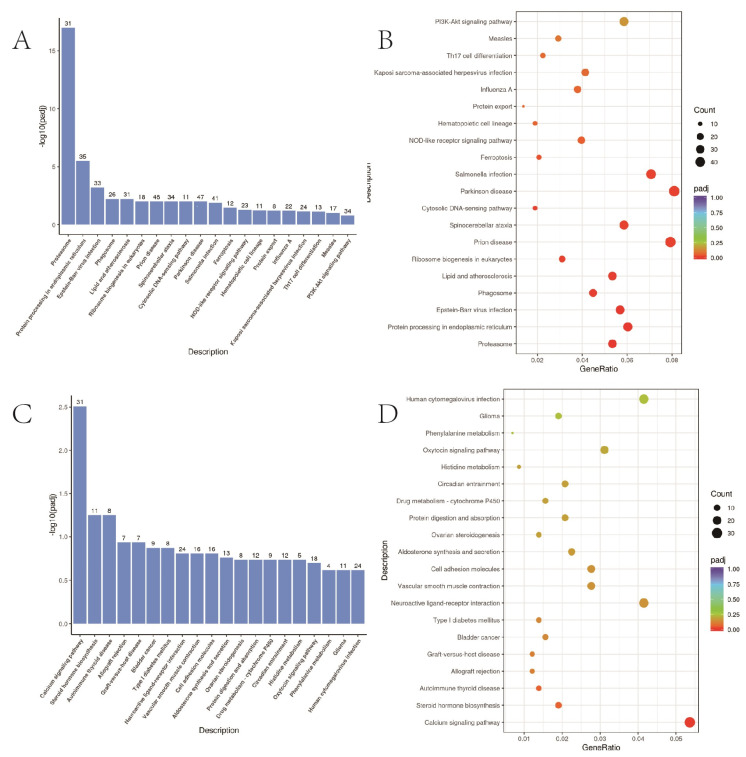
KEGG functional enrichment analysis. (**A**,**B**) A549 cells in 2D and 3D groups up-regulated the first 20 items of differential gene KEGG pathway enrichment; (**C**,**D**) A549 cells in 2D and 3D groups down-regulated the first 20 items of differential gene KEGG pathway enrichment.

**Figure 8 bioengineering-10-00667-f008:**
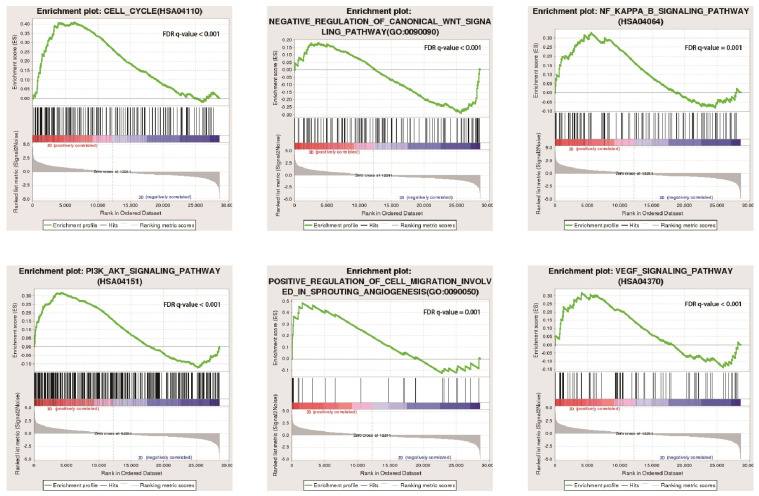
Functional enrichment analysis of A549 cells cultured in 2D and 3D.

## Data Availability

Due to the nature of this research, participants did not agree for their data to be shared publicly.

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
