# Peer review of "Characterization of 3D-Bioprinted In Vitro Lung Cancer Models Using RNA-Sequencing Techniques"

_bioengineering, 2023, doi:10.3390/bioengineering10060667_

Round 1

Reviewer 1 Report

Comments:

1.       The authors may consider changing the title to “Characterization of 3D bioprinted in-vitro lung cancer models using RNA-sequencing techniques”.

2.       The authors mentioned that “3D bioprinting technology has attracted much attention due to its great potential in tissue engineering production” – the authors should provide more information on the “great potential”. The authors can refer to some of the highly-cited review papers on bioprinting and list down the advantages of 3D bioprinting (such as controlling the spatial arrangement of cells and biomaterials, manipulating microstructure, scalable and reproducible fabrication of patient-specific tissue engineered constructs) for tissue engineering.

a.       "Print me an organ! Why we are not there yet." Progress in Polymer Science 97 (2019): 101145.

b.       "Progress in 3D bioprinting technology for tissue/organ regenerative engineering." Biomaterials 226 (2020): 119536.

3.       The authors should discuss the different 3D bioprinting approaches with relevant references based on ASTM standards, which includes extrusion, jetting and vat photopolymerization-based bioprinting. The authors can refer to the following papers on the different bioprinting approaches.

a.       Extrusion-based

                                                               i.      "Current advances and future perspectives in extrusion-based bioprinting." Biomaterials 76 (2016): 321-343.

b.       Jetting-based

                                                               i.      "Controlling droplet impact velocity and droplet volume: Key factors to achieving high cell viability in sub-nanoliter droplet-based bioprinting." International Journal of Bioprinting 8, no. 1 (2022) 424

c.       Vat photopolymerization-based

                                                               i.      "Vat polymerization-based bioprinting—Process, materials, applications and regulatory challenges." Biofabrication 12, no. 2 (2020): 022001.

4.       What is the current state of the arts for lung bioprinting? The authors should discuss what has been done for lung bioprinting and how is this work different from the previous works?

a.       "Engineering an in vitro air-blood barrier by 3D bioprinting." Scientific reports 5, no. 1 (2015): 1-8.

b.       "Fabrication and characterization of 3D bioprinted triple-layered human alveolar lung models." International Journal of Bioprinting 7, no. 2 (2021) 332.

c.       "3D Bioprinting for Regenerating COVID-19-Mediated Irreversibly Damaged Lung Tissue." International Journal of Bioprinting 8, no. 4 (2022) 616.

5.       In this work, only A549 cells (epithelial cells) were used to fabricate “lung model”. Hence, the authors need to state clearly what is the novel of this work? Different types of lung cells (epithelial, endothelial and fibroblasts) were used in other lung bioprinting papers, why is a single type of A549 cells sufficient to replicate the lung models?

6.        What is the histology of the “matured” lung models? It is critical to form a uniform layer of epithelial cells and more in-depth characterization should be performed.   

The quality of English language is acceptable. 

Author Response

Dear Professor,

Thank you very much for your letter and the referee's report. According to your comments and requirements, we have made a lot of revisions to the original manuscript. Here we enclose the revised manuscript. The revised manuscript. Corrections marked in red are attached for checking/editing purposes. If you have any questions, please do not hesitate to contact us.

Reviewer 2 Report

This manuscript presents a strategy to use extrusion-based bioprinting to build lung cancer models. Despite the importance of its topic, and the scientifically sound methodology, this work needs much improvement to reach its audience. 

Major concerns: 

1. The Introduction needs a thorough revision to provide up-to-date arguments for the importance of the present work. The literature on bioprinted models of the tumor microenvironment (TME) is vast. The field is a decade old by now, and progress has been made in many respects. It is important to clarify the positioning of this work with respect to other studies dealing with bioprinted cancer models. At present, Section 1 is a generic introduction into the paper's topic (cancer, bioprinting, hydrogel-based bioinks), but lacks focus on the state-of-the-art. 

2. The bibliography of this paper is mainly outdated and of little relevance. Several seminal works are omitted by the authors. Here is an incomplete list (none of them is ours): 

[1] Zhang YS, Duchamp M, Oklu R, Ellisen LW, Langer R, Khademhosseini A. Bioprinting the Cancer Microenvironment. ACS biomaterials science & engineering. 2016;2(10):1710-21.

[2] Zhao Y, Yao R, Ouyang L, Ding H, Zhang T, Zhang K, et al. Three-dimensional printing of Hela cells for cervical tumor model in vitro. Biofabrication. 2014;6(3):035001.

[3] Mao S, Pang Y, Liu T, Shao Y, He J, Yang H, et al. Bioprinting of in vitro tumor models for personalized cancer treatment: a review. Biofabrication. 2020;12(4):042001.

[4] Bae J, Han S, Park S. Recent Advances in 3D Bioprinted Tumor Microenvironment. BioChip Journal. 2020;14(2):137-47.

[5] Datta P, Dey M, Ataie Z, Unutmaz D, Ozbolat IT. 3D bioprinting for reconstituting the cancer microenvironment. Precision Oncology. 2020;4:18.

[6] Kang Y, Datta P, Shanmughapriya S, Ozbolat IT. 3D Bioprinting of Tumor Models for Cancer Research. ACS Applied Bio Materials. 2020;3(9):5552-73.

[7] Augustine R, Kalva SN, Ahmad R, Zahid AA, Hasan S, Nayeem A, et al. 3D Bioprinted cancer models: Revolutionizing personalized cancer therapy. Translational Oncology. 2021;14(4):101015.

[8] Meng F, Meyer CM, Joung D, Vallera DA, McAlpine MC, Panoskaltsis-Mortari A. 3D Bioprinted In Vitro Metastatic Models via Reconstruction of Tumor Microenvironments. 2019;31(10):e1806899.

A focused literature search is needed to boost the credibility of this paper.  

3. In Section 3 (Results) the annotation of several figures is too small. Figure 7, for instance, is impossible to read even if magnified because it is of low resolution. One option is to select illustrations of reasonable size and present extra details in the Supplementary Material. 

4. The Discussion should focus on the analysis of the reported results in the context of the findings of other groups. In the above list, ref. [8], (Meng et al., 2019), deals with lung cancer cells from the same cell line, A549. Also, ref. [2], (Zhao et al., 2014), relies on the same bioprinting approach. All these works could have been discussed in Section 4. Instead, the authors decided to present a historical account dating back to the early 1990s, and restate certain results without literature citations (except for the last paragraph, which would actually fit in Section 5).

5. This manuscript is plagued by several imprecise statements, such as "the 3D bio printed hydrogel has a porous mechanism" (line 62), "the printed hydrogel can co-culture multiple cells" (line 64), "in 3D bioprinting, gelatin mainly serves as the extracellular matrix of cells" (line 83). A careful revision of the entire text is needed to eliminate awkward language and, instead, use the professional jargon of 3D bioprinting; see 

[9] Moroni L, Boland T, Burdick JA, De Maria C, Derby B, Forgacs G, et al. Biofabrication: A Guide to Technology and Terminology. Trends in Biotechnology. 2018;36(4):384-402.; 

[10] Groll J, Boland T, Blunk T, Burdick JA, Cho D-W, Dalton PD, et al. Biofabrication: reappraising the definition of an evolving field. Biofabrication. 2016;8(1):013001; and 

[11] Groll J, Burdick JA, Cho DW, Derby B, Gelinsky M, Heilshorn SC, et al. A definition of bioinks and their distinction from biomaterial inks. Biofabrication. 2018;11(1):013001.  

Minor remarks: 

a. Please check the punctuation. In several instances a space is lacking after the period (ex. lines 57, 60).   

b. I would avoid referring to "this part of the study" (lines 27, 319, 345, 349, 368) because it suggests that the paper does not describe the entire study, just part of it.  

c. There is no need for Bold characters on lines 229-230. 

d. Instead of "tumor cell balls" (line 251) I would prefer the term "tumor cell aggregates". 

e. Instead of "volcanic maps" (line 275) please use "volcano maps".

While describing the experimental results, please stick to past tense instead of a mix of past tense and present tense (e.g. lines 222, 223). 

Author Response

(The authors gave the same response as above.)

Reviewer 3 Report

â—‹     This manuscript aimed to construct the lung cancer model and evaluated the model using transcriptome sequencing. However, this manuscript has some concerns for the further proceeding of this journal as below:

l  The manuscript needs to be revised in terms of the overall story as it currently focuses on individual results rather than presenting a cohesive narrative, with a need to reconsider the configuration of the figure outline and manuscript flow. For example, Figure 4 and 5, both results indicates morphological observation of cancer cells. Thus, it can be combined as a one figure and it would help to increase reading comprehension.

l  In introduction, it is recommended to describe the advantages and requirements of the 3D bioprinting in detail and refer the related studies as below.

-       Kim, Byoung Soo, et al. "Decellularized extracellular matrix-based bioinks for engineering tissue-and organ-specific microenvironments." Chemical Reviews 120.19 (2020): 10608-10661.

-       Jorgensen, Adam M., James J. Yoo, and Anthony Atala. "Solid organ bioprinting: strategies to achieve organ function." Chemical reviews 120.19 (2020): 11093-11127.

l  Figure 1, it is recommended to increase the size of the words to read clearly.

l  Scale bars should be modified in all figures.

l  Line 229 on page 6, the bold texts should be modified.

l  Line 237 on page 7, it need to rearrange the formation of the sentence.

l  Figure 3A-C, it would be better to make a border line of the printed structure to recognize the location of dead cells easily. Additionally, it should be mark the scale bar, same as figure 3D.

l  Figure 4 and 5, it would be better to show the quantitative data to understand the results easier.

 Extensive or moderate editing of English language required

Author Response

(The authors gave the same response as above.)

Round 2

Reviewer 1 Report

The authors have addressed all the comments, the revised manuscript can be accepted in present form 

Author Response

Thank you very much for reviewing our paper again. Wish you a happy life. If you have any questions, please contact us in time.

Reviewer 2 Report

This manuscript has been revised by taking into account part of my previous report. Nevertheless, my main concerns were not addressed properly. The remaining problems include the following: 

1. The use of the professional language of 3D bioprinting and biofabrication would have required a thorough revision of the entire text, not just the addition of a few new paragraphs. 

2. The Introduction, as it stands, is a mini-review of what has been done before, but it does not argue the need for the present study. To be fair, I need to admit that some of the references suggested in my first review have been commented in new paragraphs. An Introduction, however, has three main roles: (i) to provide the background information needed by the reader to grasp the original results, (ii) to argue the importance of the proposed research, and (iii) to explain the objectives of the study pointing out the knowledge gap addressed by it. An unfocused review of previous research is not effective from these points of view. 

 3. The most important weakness of this paper is that it is still disconnected from the literature, mainly because the Discussion continues the same strategy of providing a literature review that made the Introduction unfocused. The Discussion is supposed to analyze the relevance of the reported results by  (detailed, specific) comparison with other publications. It is not enough to state that another group's finding "was the same as the result of this experiment" (line 363). What was similar? What was different? What is the importance of these results? Also, vague language makes a single-sentence summary of an entire paper a confusing experience for the reader. For example, the sentence "Su Su Htwe et al.[50] found that the activation degree of NF-κB in 2D cells was much higher than that in 3D cells" (line 379) has left me wondering what the authors mean by "2D cells" and "3D cells", and how do their specific culture conditions compare to those of reference [50]. 

In conclusion, this manuscript is still far from being informative, and I can only hope that the authors will have the willingness and the ability to make it so.   

The language of this manuscript is comprehensible, but vague and unprofessional in many instances. The improvement compared to the previous version is unsatisfactory. 

Author Response

According to your comments and requirements, we have made a lot of revisions to the original manuscript. If you have any questions, please contact us immediately. Please see the attachment.

Reviewer 3 Report

â—‹     This revised manuscript has been clearly improved by addressing the previous comments. In this version, it gives more comprehensive descriptions. Therefore, it is recommended to consider this manuscript as a publication in this journal.

Minor editing of English language required

Author Response

(The authors gave the same response as above.)
